

# E3SM Chemistry Diagnostics Package (ChemDyg) Version 0.1.4

6  Hsiang-He Lee[*1], Qi Tang[1], and Michael J. Prather[2]

7  [1]Lawrence Livermore National Laboratory, Livermore, CA, U.S.A.

8  [2]University of California, Irvine, CA, U.S.A.

18  *Corresponding author address: Dr. Hsiang-He Lee, 7000 East Avenue, Livermore, CA, 94550,

19  U.S.A.

20  E-mail: lee1061@llnl.gov





## Abstract

The E3SM Chemistry diagnostics package (ChemDyg) is an open-source software, all diagnostic scripts written in Python, developed to support the Department of Energy (DOE) Energy Exascale Earth System Model (E3SM). The current version 0.1.4 of ChemDyg generates several diagnostic plots and tables for model-to-model and model-to-observation comparison, including 2-dimential contour mapping plots, diurnal and annual cycle plots, time-series plots, and comprehensive processing tables. ChemDyg is executed by zppy, which is a post-processing toolchain for E3SM written in Python. The ChemDyg codebase is designed to be modular, and each diagnostics set is self-contained. Each set has its own driving script that includes set-specific file input/output and a main python script for calculation and plotting. The outputs from each diagnostics set, including figures and tables, are organized in the main HTML page to make it interactive through a browser.

This paper is a comprehensive description of E3SM ChemDyg (as of version 0.1.4) including the details of each diagnostics set and its required input data formats. This tool has enough flexibility for future ChemDyg developers to increase the addition of new observational datasets and new diagnostics sets.





# 1 Introduction

The U.S. Department of Energy (DOE) Energy Exascale Earth System Model (E3SM) version one (E3SMv1) (Golaz et al., 2019) was first released in 2018 to feed the DOE mission needs for producing robust actionable predictions of Earth system variability and change, with an emphasis on the most critical scientific questions facing the nation and DOE (Leung et al., 2020). E3SM version two (E3SMv2) (Golaz et al., 2022) released in 2021 is based on E3SMv1 with significant improvements in existing and emerging architectures and some key climate physics. The atmosphere component of E3SMv2, E3SM Atmosphere Model (EAMv2), improved the dynamic core to solve the equations of motion in a rotating reference farm with the hydrostatic and shallow atmosphere approximations (Taylor et al., 2020). Some physics schemes have been updated in EAMv2 as well to improve computational performance. For example, the internal call order and array structure of the Cloud Layers Unified By Binormals (CLUBB) (Golaz et al., 2002; Larson, 2017) have been changed to permit contiguous memory access and also to eliminate the unrealistic pockets of supersaturation that were resolved in the microphysics scheme in EAMv1. The Zhange-McFarlane deep convection scheme (Zhang and Mcfarlane, 1995) adopted two features to improve its simulated diurnal cycle precipitation (Xie et al., 2019). The upcoming version three of E3SM (E3SMv3) has interactive chemistry which includes the prognostic linearized ozone chemistry scheme version three (Linoz v3) (Prather and Hsu, 2010; Hsu and Prather, 2010) and the University of California, Irvine (UCI) chemistry mechanism (chemUCI).

The atmospheric chemistry in EAMv1 was the Ozone model version one (O3v1) which is prescribed tropospheric ozone based on decadal monthly zonal mean latitude-pressure data from the input4MIPS Ozone dataset v1.0 (Hegglin et al., 2016) and prognostic stratospheric ozone by the linearized chemistry version two (Linoz v2) (Hsu and Prather, 2009). O3v1 calculated





stratospheric ozone interactively with Linoz v2, but it can result in unrealistic ozone distribution
when the simulated tropopause was higher than that of the prescribed data. The model tends to
assign stratospheric ozone abundance to the tropospheric model grid boxes. In EAMv2, the O3v2
model (Tang et al., 2021) was implemented to overcome the issues in the O3v1 model and to
perform a more comprehensive evaluation of the ozone simulation. O3v2 is able to interact with
the tropopause changes and hence captures the naturally sharp ozone cross-tropopause gradient.
Furthermore, the stratosphere-troposphere exchange (STE) flux of ozone, which is an important
tropospheric ozone budget term, is available to diagnose in O3v2 due to the new method of ozone
sink at the lower boundary in O3v2. In the upcoming EAMv3 release, we update the stratospheric
chemistry package from Linoz v2 to Linoz v3 along with the new interactive tropospheric
chemistry (chemUCI). Linoz v3 extends the Linoz v2 capabilities to include more tracers
important for the E3SM goals, including prognostic ozone, methane ($CH_4$), nitrous oxide ($N_2O$),
and reactive nitrogen compounds ($NO_y$) as well as diagnostic stratospheric water vapor. chemUCI
consists of 28 advected tracers for the $O_3$-$CH_4$-$HO_x$-$NO_x$-NMVOCs chemistry in the troposphere.
chemUCI originated from the University of California, Irvine (UCI) chemistry transport model
(CTM)'s ASAD package (Carver et al., 1997; Wild et al., 2003).
The current model evaluation packages have comprehensive diagnostic sets to evaluate the
meteorological and climatic features (e.g., temperature, precipitation, radiative effects, etc.). For
example, the E3SM Diagnostics Package (E3SM Diags) is a Python-based Earth System Model
evaluation tool, which was developed to support E3SM development. The core set of E3SM Diags
is the seasonal and annual mean physical climate for the major variables. The plot sets cover
latitude-longitude maps, maps focusing on the polar regions, Pressure-Latitude zonal mean contour
plots, etc. Compared to the version two of E3SM Diags, the current version 2.7 of E3SM Diags





(Zhang et al., 2022) includes some new diagnostics sets for Quasi-biennial Oscillation, El Nino-
Southern Oscillation, streamflow, diurnal cycle of precipitation, analysis from ARM ground-based
facilities, tropical cyclone, stratospheric column ozone (SCO) and tropospheric column ozone
(TCO).

As mentioned before, O3v2 in EAMv2 has a big improvement to overcome the issue of the

vertical distribution of ozone abundance, especially near the tropopause, in O3v1 because the issue
can result in significant impacts on the radiation transfer in E3SM.  In EAMv3, the model has fully
interactive chemistry in both the troposphere and stratosphere.  The ozone-only diagnostics in
E3SM Diags v2.7 cannot meet the EAMv3 chemistry evaluation requirements.  Thus, a
systematized analysis tool for E3SM chemistry is necessary for E3SM model developers.

This paper introduces a new Python package for E3SM chemistry diagnostics: ChemDyg,

which has been developed to support E3SMv3 chemistry development.  The paper is a
comprehensive description of ChemDyg (as of version 0.1.4) including the details of each
diagnostics set.  An overall description of ChemDyg, code structure, associated input data, and
processing of model output are given in Section 2.  The detail of each diagnostics set is shown in
Section 3.  A discussion of future applications and a summary are provided in Section 4.

## 2   The overview of ChemDyg

### 2.1   Code structure and workflow

ChemDyg is an open-source software, and all diagnostic scripts are written in Python.

ChemDyg is executed by zppy, which is a post-processing toolchain for E3SM
(https://e3sm.org/resources/tools/end-to-end-processing/zppy/).  A configure run script (i.e., .cfg
file) in zppy controls the tasks in ChemDyg.  Users have high flexibility to specify the setting at





multiple levels (i.e., for general input and output information, and different time period for each
sub task) in the configure run script. Users can even select fewer specific sub tasks if necessary.
Some diagnostics sets in ChemDyg request reformatted E3SM outputs (e.g., remapping,
seasonal/annual mean climatology, etc.), instead of default monthly data with a native grid. The
post-processing E3SM output can also be handled by zppy for running netCDF operators
(http://research.jisao.washington.edu/data_sets/nco/) (see Section 2.3 for details). Observations
and reanalysis data used in some diagnostics sets of ChemDyg are pre-processed, and the reference
path is default assigned corresponding to different DOE machines. Figure 1 depicts a schematic
overview of the code structure and workflow.
The ChemDyg codebase is designed to be modular, and each diagnostics set is self-
contained. Each set has its own driving script that includes set-specific file input/output and a
main Python script for calculation and plotting. The outputs from each diagnostic set, including
figures and tables, are organized in the main HTML page to make it interactive through a browser.
## 2.2   Data involved for comparison
### 2.2.1   Observations of surface ozone
Analyzed 10 years (2000-2009) hourly surface ozone ($O_3$) data from air quality networks
in North America (NA; 25-49°N and 125-67°W) and Europe (EU; 36-71°N and 11°W-34°E) is
adopted from Schnell et al. (2015). The data sets used for NA are mainly from three networks: the
United States (US) Environmental Protection Agency's (EPA) Air Quality System (1633 stations;
http://www.epa.gov/ttn/airs/aqsdatamart), the US EPA's Clean Air Status and Trends Network (92
stations; http://epa.gov/castnet/javaweb/index.html), and Environment Canada's National Air
Pollution Surveillance Program (207 stations; http://maps-cartes.ec.gc.ca/rnspa-
naps/data.aspx?lang=en). The observational data sets in Europe (EU) are provided by the





European Environment Agency's air quality database (2123 stations;
http://www.eea.europa.eu/data-and-maps/data/) and the European Monitoring and Evaluation
Programme (162 stations; Hjellbrekke et al. (2013)).
In order to have a commensurate comparison between station measurements and model
simulations, the surface observed $O_3$ is interpolated onto the $1° \times 1°$ hourly grid-cell averaged $O_3$
data over NA and EU by using the interpolation scheme described in Schnell et al. (2014). The
maximum daily 8-hour averages (MDA8) are interpolated from the hourly measurements and
subsequently derive the MDA8 at each grid cell (Schnell et al., 2015).
2.2.2 Surface ozone from ACCMIP and UCI CTM
The eight models in the Atmospheric Chemistry and Climate Model Intercomparison
Project (ACCMIP; Lamarque et al. (2013)) with archived hourly surface $O_3$ are used for surface
$O_3$ comparison in ChemDyg. The analyzed data in the plots are adopted from Schnell et al. (2015).
Most ACCMIP models provide 10 years of data, starting in either model year 2000 or 2001, to
closely align with observations. Table 1 provides a summary and references of the models used in
this comparison. Besides the ACCMIP models, Schnell et al. (2015) also used one hindcast
simulation over the same period as the observations from the University of California Irvine
Chemical Transport Model (UCI CTM) (Holmes et al., 2013).
For an appropriate comparison between the models and measurements, the modeled hourly
$O_3$ abundances (mostly at 2° to 3° resolution; Table 1) are regridded to the same $1° \times 1°$ cells as
the observations using first-order conservative mapping.





### 2.2.3 Tropospheric ozone from CMIP6
The 3-dimensional $O_3$ from the five models in Phase 6 of the Coupled Model
Intercomparison Project (CMIP6) Historical experiments are used in ChemDyg for tropospheric
column ozone (TCO) comparison. We use the archived $O_3$ data from AERmon (CMIP6 table for
monthly atmospheric chemistry and aerosol data) to derive the tropospheric $O_3$ burden on the
model grids, along with the tropopause using the World Meteorological Organization (WMO)
definition of tropopause on 3-dimension temperature. The selected models are based on Griffiths
et al. (2021). The $O_3$ data are available for from 1850 to 2014, and a short summary of CMIP6
models is presented in Table 2.
### 2.2.4 Observations for ozone hole evaluation
The observational metrics used for $O_3$ hole evaluation are daily geographically resolved
total column ozone (TOZ), which are collected from three satellite remote sensing data: the Total
Ozone Mapping Spectrometer (TMOS) on the NASA/NOAA Nimbus-7 satellite (Mcpeters et al.,
1996), the Ozone Monitoring Instrument (OMI) on the Aura satellite (Schoeberl et al., 2004), and
the Ozone Mapper and Profiler Suite (OMPS) on NASA's Suomi National Polar-orbiting
Partnership (NPP) satellite (Flynn et al., 2014). Based on these daily TOZ data, the NASA Ozone
Watch website (https://ozonewatch.gsfc.nasa.gov) compiles the daily records of the Antarctic
ozone hole area defined as TOZ < 220 DU (Dobson units, milli-cm-amagats) and minimum TOZ
in the Southern Hemisphere (SH). The data obtained from the Ozone Watch website are used in
ChemDyg to evaluate model simulations.





2.2.5   NOAA surface carbon monoxide

The observed surface carbon monoxide (CO) is from the Global Greenhouse Gas

Reference Network for the Carbon Cycle and Greenhouse Gases (CCGG) Group, which is part of
NOAA's Global Monitoring Laboratory (GML) in Boulder, Colorado (Petron et al., 2022).  The
Reference Network measures four greenhouse gases which are the main long-term drivers of
climate change and important indicators of air pollution: carbon dioxide ($CO_2$), methane ($CH_4$),
nitrous oxide ($N_2O$), and CO.  The Reference Network measurement program collects data mainly
from in-situ measurements at four baseline background sites and 8 tall towers.  Besides long-term
in-situ observations, flask-air samples are collected by volunteers at over 50 additional regional
background sites and from small aircraft.  The observed CO metrics used in ChemDyg are flask-
air                                      samples                                      (see
https://gml.noaa.gov/aftp/data/trace_gases/co/flask/surface/README_surface_flask_co.html  for
details) collected from 12 stations (Table 3) for model evaluation, and the observational data cover
the period from 1988 to 2021.
2.3   Pre-processing of model output

Due to the purpose of each chemistry diagnostics set, besides the default monthly mean

E3SM output (h0 files), three types of time frequency output are required for ChemDyg: monthly
(0 in `nhtfrq`) instantaneous ('I' in `avgflag_pertape`) output (for example, h1 files), hourly (-1
in `nhtfrq`) instantaneous output (for example, h2 files), and daily (-24 in `nhtfrq`) averaged ('A'
in `avgflag_pertape`) output (for example, h3 files).  Appendix A shows an example setup of
E3SM run script to generate these three types of time frequency output in the coordinated output
files (h1~h3) with the associated variables.  Note that `history_gaschmbudget_num` should be the





same as the monthly instantaneous output (here is h1 in Appendix A). The E3SM filenames can
be changed according to user preference and specified in the ChemDyg run script.

The pre-processing tasks include generating regridded climatology from the monthly

output, and regridded monthly/hourly time-series files, as input for various diagnostics sets in
ChemDyg. The default format of E3SM output is on a native grid. In order to have a
commensurate comparison to observations or for the plotting purpose, the model outputs need to
regrid to different resolutions. The most common one is to interrelate to a $1° \times 1°$ grid as the
CMIP6 format. As we mentioned in Section 2.2.2, the metrics for surface $O_3$ comparison are
required to be remapped to the same $1° \times 1°$ cells as the observations but covering the selected
area in NA and EU only.

## 3   Examples of diagnostics


Version 0.1.4 of ChemDyg generates 11 types of plots and four types of tables for model-

to-model and model-to-observation comparison, including 2-dimensional contour mapping plots,
diurnal and annual cycle plots, time-series plots, and comprehensive processing tables. Table 4
shows a summary of the basic sets of diagnostics in ChemDyg, including a short description,
supported model input format (i.e., native/regarded grid and time-series/climatology), and
associated observations/reanalysis data. This section shows some examples to illustrate the
application of this diagnostics package in evaluating E3SM chemistry. Note that all diagnostics
figures/tables included in this paper were extracted from several ChemDyg runs which are
simulated from E3SMv3 testbase candidates for demonstration purposes.





## 3.1 Pressure-Latitude plots

Ozone abundance and distribution is one of the main focuses of E3SMv3 chemistry development. This diagnostics set includes Pressure-Latitude zonal mean contour plots of annual mean $O_3$, $O_3$ in troposphere only ($O_3$ Trop), specific humidity (Q), and temperature (T). These figures provide the overall of $O_3$ vertical profile associated with potential temperature gradient and other related parameters (i.e., Q and T). Figure 2 demonstrates a Pressure-Latitude plot of multi-year zonal mean $O_3$ abundance and potential temperature. We also present the zonal mean $O_3$ profile for the standard E3SMv2 simulation and their differences to evaluate the $O_3$ implementation in E3SMv3.

E3SMv2 uses the World Meteorological Organization (WMO) defined tropopause, which is the lowest level at which the temperature lapse rate decreases to 2 K per km or less, with the average lapse rate between this level and all higher levels within 2 km. Based on this definition, the model is hard to illustrate a folded tropopause due to a frontal system or other 3-dimensional (3D) tropopause structure. Thus, in E3SMv3 chemistry, we introduce an artificial tracer called E90 to define a tropopause that effectively separates stratospheric air from tropospheric air from a chemical composition perspective (Prather et al., 2011). This diagnostics set also shows the WMO defined tropopause (the yellow line in Fig. 2a and the cyan line in Fig. 2b)) and E90 defined 3D tropopause (the magenta line in Fig. 2a) side by side.

## 3.2 Latitude-Longitude plots

Meteorological factors affect the chemical processes and alter chemistry concentrations, including $O_3$. In a recent study (Sitnov et al., 2017), precipitable water vapor and total column $O_3$ have strong connections with the North Atlantic Oscillation. Thus, we highlight total precipitable water (TMQ) (Fig. 3) in ChemDyg.





### 234     3.3    Nitrogen oxides ($NO_x$) emission plots

Nitrogen oxides ($NO_x$) emissions have been a constant concern because of their role in air

pollution leading to smog and acidic wet and dry deposition. $NO_x$ are also important in affecting
global $O_3$ concentrations through their chemical reactions with hydrocarbons and then as $O_3$
precursors. The largest sources of $NO_x$ in the troposphere are fossil fuel combustion, biomass
burning, lightning discharges, aircraft emissions, and microbial activity in soils through
biogeochemistry processes. This diagnostics set mainly focuses on lighting discharges and aircraft
emissions (Fig. 4).

### 242     3.4    Tropospheric column ozone (TCO) comparison

Ozone is a short-lived reactive gas and an oxidizing species with adverse effects on human

health (Lippmann, 1991). Tropospheric $O_3$ production is from the photochemical oxidation of CO,
$CH_4$, and non-methane volatile organic compounds (NMVOCs) in the presence of nitrogen oxides
(i.e., NO and $NO_2$). Thus, the TCO burden is controlled by the balance between chemical
production and loss processes (Section 3.11). The $O_3$ production and loss vary between models
due to different approaches in representing the processes involved and different budget terms
defined. The definition of tropopause also affects the diagnosed TCO burden and any influx from
the stratosphere. Based on Griffiths et al. (2021), we examine TCO for five CMIP6 models (Table
2) which are derived from the archived $O_3$ and tropopause data from AERmon. This diagnostics
set is designed to compare the time series of global mean TCO between E3SM and the CMIP6
models (Fig. 5).

### 254     3.5    Surface ozone diurnal and annual cycle comparison

Following the method in Schnell et al. (2015) using air quality networks to evaluate

modeled surface $O_3$, 10 years (2000-2009) hourly surface $O_3$ measurements from air quality





networks in NA and EU and the ACCMIP model simulated surface $O_3$ (Table 1) are used in this
diagnostics set to evaluate E3SM surface $O_3$ performance. Because of different chemistry regions,
the analysis domain in NA is split into western (WNA) and eastern (ENA) regions at 96°W, and
in EU is split into southern (SEU) and northern (NEU) regions at 53°N.
The diagnostics set of surface $O_3$ diurnal provides diurnal cycles of hourly $O_3$ abundance
averaged over winter (DJF) shown in Fig. 6a and summer (JJA). The set of annual cycles is
particularly for MDA8 $O_3$ abundance (Fig. 6b).
3.6  Surface CO comparison
As we mentioned in Section 3.1, TCO production is from the photochemical oxidation of
CO and other chemical species. CO is one of the main precursors of $O_3$ production. Compared to
$O_3$, the lifetime of CO is longer, about two months, to make CO a good indicator for regional air
quality. This diagnostics set uses NOAA surface CO collected from 12 stations (Table 3) for
E3SM model evaluation. Figure 7 shows the time series of surface CO burden from observation
(red solid line) and E3SM (black solid line) at the BRW observational site. We also calculate the
mean, anomaly, and tendency of surface CO burden during the analyzed period.
3.7  Ozone hole
Antarctic ozone hole commonly is defined by three types of metrics: the area of the hole
(adding the areas below the same threshold in the total column ozone (TOZ) field), the minimum
TOZ value within the hole, and the Antarctic ozone mass deficit (Uchino et al., 1999; Huck et al.,
2007). Since 1990, the evolution of the Antarctic ozone hole has been driven primarily by
dynamical variation due to the chlorine levels driving ozone depletion inside the Antarctic winter
vortex. Several previous studies have calculated indicators of Antarctic ozone depletion and





Uchino et al. (1999) examined the changes in the area of the Antarctic ozone hole using a threshold
of TOZ less than 220 DU and the ozone mass deficiency within the ozone hole.

In this diagnostics set, we use the daily ozone hole diagnostics from the NASA Ozone

Watch website as a reference and two metrics: the area with less than 220 DU in TOZ and the
minimum TOZ (Fig. 8) to evaluate the model capability for capturing the ozone hole variation
compared to the observations.
3.8    Total column ozone and temperature with equivalent latitude

Bodeker et al. (2002) used the satellite data to analyze the long-term evolution of the

Antarctic ozone hole and its dependence on the size of the dynamical vortex and meridional
temperature structure within the polar vortex because the 2000 ozone hole was the largest on
record.  In their study, the vortex has strengthened from 1981 to 2000, and the average equivalent
latitude location of the center of the vortex edge has remained unchanged at ~62°S over the 20
years.  Here, equivalent latitude zonal means of these data have been calculated to show the
expansion of the Antarctic ozone hole and its encroachment on the vortex edge.  Furthermore, they
calculated daily equivalent latitude zonal mean 550 K temperature isentrope profiles and found
from Jun to August the equivalent latitude at which the temperature profiles fall below 195 K has
changed little over the 20-year period.  Thus, the increase in the size of the Antarctic ozone hole
during the period was not caused by either the increase in the size of the dynamical vortex or the
area of temperatures below 195 K.

The diagnostics set of TOZ with equivalent latitude has similar methods as Bodeker et al.

(2002) to analyze daily TOZ with equivalent latitude at 64°S to provide time series and daily
evolution of the minimum TOZ with equivalent latitude (64°S) area from 1 July to 31 December
as measured by the minimum total column ozone (Fig. 9).





The diagnostics set of temperature with equivalent latitude is to calculate the mean

temperature averaged from 1 July to 31 December at three different altitudes (i.e., 14, 20, and 25

304       km) with various equivalent latitudes from 60°S to 78°S (Figure 10). It can help us to indicate the

approximate temperature for polar stratospheric cloud formation (it was set as 197.5 K in E3SMv2

but now is 198 K in E3SMv3). Because this diagnostics set is computationally expensive, we

suggest users only turn on this set when it is necessary.

3.9    Ozone stratosphere-troposphere exchange flux

The stratosphere-troposphere exchange (STE) flux of ozone is a key budget term for

tropospheric ozone abundance. As mentioned before, in O3v2 the net STE ozone flux is estimated

from the loss in the near surface atmospheric layers (lowest four layers). This approach can limit

the global mean ozone flux based on proxy relationships with other trace gases, giving us a broad

range of 400-600 Tg $O_3$ per year (Murphy and Fahey, 1994; Mclinden et al., 2000; Olsen et al.,

2004; Olsen et al., 2001; Hsu et al., 2005). Ozone is conserved in the rest of the troposphere and

thus the STE flux is taken up by these lowest layers. It is resolved geographically and monthly

but because of the tropospheric transport from the tropopause to the lowest layers, the STE ozone

flux diagnosed this way will differ from the tropopause-crossing flux in location and with a slight

time delay of less than a month (Jacob, 1999). Figure 11 shows thee time series and mean annual

cycle in the north hemisphere (NH), south hemisphere (SH), and global.

3.10   Chemistry tendency table

The diagnostics set of chemistry tendency is designed to help E3SM model developers

pinpoint the simulation errors from potential chemistry processes when the model generates

unrealistic chemistry abundance. The tendency calculation is driven by the chemistry driver in

E3SM chemistry module from 10 processes: implicit solver (TDI), explicit solver (TDE), aerosol-





gas exchange (TDA), chemistry processes from Linoz (TDL), reset negative values (TDN), setting
upper boundary values (TDU), setting lower boundary values (TDB), surface emission (TDS), dry
deposition (TDD), and wet deposition (WD). Chemistry tendency outside the chemistry module
(e.g., dynamic driven transport) is considered in TDO. The tendency is due to the reset mixing
ratio in the stratosphere after the implicit solver and explicit solver are characterized as TRI and
TRE, respectively. Figure 12 shows a schematic workflow of chemistry tendency processes and
their associated working files.
Those tendency outputs are a column integrated 2-dimentional format for each chemistry
tracer to minimize the file size. In order to better understand the chemistry processes crossing
different vertical layers in the atmosphere, we classify the entire model column into 4 layers based
on the standard 72 model layers: top of the model to 100 hPa (L1), 100 to 267 hPa (L2), 267 to
856 hPa (L3), and 856 hPa to the surface (L4). Variables associated with the Linoz process (i.e.,
variables named O3, N2OLNZ, and CH4LNZ) have a specific output layer for the troposphere
(trop). Users can decide how to classify these 4 layers in the E3SM run script as shown in
Appendix A. This diagnostics set generates annual and seasonal chemistry tendency tables with
both HTML and text formats.
3.11 Chemistry closure check and burden table
The diagnostics set is designed for the closure check of each chemical species by
calculating Level-2 normalization relative difference between the sum of total tendencies listed in
Fig. 12 and the difference of chemistry burden (MSD) before and after the tendency calculated. A
table generated by this diagnostics set lists not only the chemistry closure check but also the mean
concentration (Tg) and volume mixing ratio (mol mol$^{-1}$) of each chemical species estimated after
dry deposition.





Note that the diagnostics set of the chemistry tendency table mentioned in Section 3.10
requires annual mean climatology data, while the diagnostics set of chemistry closure check needs
monthly time-series output. Thus, this diagnostics set requests more computational resources, and
the diagnostics period should be enough for one or two years.
3.12  Chemistry production and loss tendency table
This diagnostic table focuses on the $O_3$ production and loss in the implicit solver and the
CO production and loss in the explicit solver.
The tropospheric $O_3$ abundance is dominated by the balance between chemical production
and loss processes, deposition at the surface, and downward transport from the stratosphere. In
order to check the $O_3$ production and loss tendency in the implicit solver, we calculate each
chemical reaction for $O_3$ production and loss used in chemUCI (Appendix B) and then sum of
reactions for the production (TIP) and loss (TIL) tendency right after the implicit solver. After the
implicit solver, the chemistry module is to reset the volume mixing ratio in the stratosphere to the
mixing ratio before the implicit solver for all chemical species because the stratospheric chemistry
is handled by Linoz v3. The chemistry module also resets the diagnostic reaction rate in the
stratosphere to zero. The tendency due to the stratospheric volume mixing ratio reset after the
implicit solver is identified as TRI in the tendency table. After the stratospheric volume mixing
ratio reset, we calculate the sum of chemical reactions for the $O_3$ production (CIP) and loss (CIL)
again for a final chemistry closure check after the implicit solver and the stratospheric volume
mixing ratio reset. Figure 13 shows a schematic workflow of chemistry tendency processes for
the $O_3$ production and loss diagnostics in the implicit solver and after the stratospheric reset.
Current chemical reactions for the $O_3$ production and loss used in chemUCI (Appendix
B) miss some important reactions with NO and $NO_2$. We also calculate the chemistry reactions





suggested by Prof. Michael Prather (see in Appendix B) for the $O_3$ production and loss (MPP and
MPL) in the tendency diagnostic table for comparison.

The mixing ratio tendency of CO is handled by the explicit solver (TDE). The production

(TEP) and loss (TEL) tendency are calculated in one subroutine (i.e., `exp_prod_loss`) and the
code structure is relatively straightforward compared to the implicit solver. The tendency due to
the stratospheric volume mixing ratio reset after the explicit solver is set as TRE in the tendency
table. After the stratospheric diagnostic reaction reset (important in the explicit solver), we call
the subroutine again to calculate the CO production (CEP) and loss (CEL) tendency for a final
chemistry closure check after the explicit solver and the stratospheric volume mixing ratio reset.
3.13  Chemistry high-level summary table

The chemistry high-level summary table provides a short summary for $O_3$, CO, and $NO_x$.

The summary table for $O_3$ includes burden, deposition, production, loss, net change from
production and loss, TCO, SCO, and STE. Those are introduced in other diagnostics sets in the
previous sections. The summary table for CO includes burden, emission, deposition, production,
loss, and net change from production and loss. Regarding $NO_x$, the summary table shows their
total burden, emission, and deposition. Once NOx emission variables from lighting (NO_TDLgt)
and aircraft (NO2_TDAcf) are available in the E3SM outputs, they will be presented in the
summary table as well.
## 4   Discussion and summery

This paper introduces a new Python package for E3SM chemistry diagnostic - ChemDyg,

which is developed to support E3SMv3 chemistry development. The chemistry tendency table is
generated from the diagnostics set designed for E3SM developers to identify the chemistry
processes for unrealistic chemistry abundance. Because the diagnostics set uses the annual and



seasonal mean climatology input data, it is easy to overlook the errors from the time step or diurnal

scale. Thus, we provide one stand-alone Python script, called `DEBUG_chem_diags_timestep.py`

(`https://github.com/E3SM-`

`Project/ChemDyg/blob/main/DEBUG_chem_diags_timestep.py`) to generate a chemistry

tendency table for each time step, hourly or daily basis depends on the time frequency of output

file.

A completed ChemDyg diagnostics requires extra chemistry outputs compared to other

E3SM diagnostics packages. In order to reduce the maximum number of E3SM outputs but keep

essential variables for ChemDyg, we designed a simplified version of ChemDyg to remove the

outputs for the chemistry tendency table and the chemistry closure check table (roughly reducing

70% of full diagnostics outputs). The users only need to turn on the flag of

`history_chemdyg_summary` in the E3SM run script and turn off other diagnostic flags (i.e.,

`history_gaschmbudget_2D`, `history_gaschmbudget_2D_levels`,

`history_UCIgaschmbudget_2D`, and `history_UCIgaschmbudget_2D_levels` in Appendix A).

The current released version (v0.1.4) of ChemDyg generates 11 types of plots and 4 types

of tables for model-to-model and model-to-observation comparison (Table 4), mainly for E3SM

chemistry. The diagnostic package includes 2-dimentional contour mapping plots, diurnal and

annual cycle plots, time-series plots, and comprehensive processing tables. The upcoming version

0.1.5 of ChemDyg will not only generate plots but also their corresponding data in the NetCDF

format. The users only need to turn on the flag, ncfile_save = 'true', in the .cfg run script.

ChemDyg will continue to develop as one of the main evaluation packages for component models

of E3SM, focusing on more chemistry species in future development.

Regarding technical enhancements, some diagnostics sets request more nodes and longer

computational time, especially for the table of chemistry closure check and the plot of temperature



with equivalent latitudes. Further improvement is needed to solve I/O and computational
limitations of zeep (walltime and the number of nodes).

## Appendix A: E3SM output requirement

This section provides an example setup to generate necessary variables for ChemDyg when

running E3SM on DOE supercomputers. Note that the paths of input data might change on
different supercomputers.

```
cat << EOF >> user_nl_eam
nhtfrq = 0,0,-1,-24
mfilt = 1,1,240,30
avgflag_pertape = 'A','I','I','A'
history_gaschmbudget_num = 2
fincl1='E90','N2OLNZ','NOYLNZ','H2OLNZ','CH4LNZ', 'TOZ',
'O3','OH','HO2','H2O2','NO','NO2','NO3','N2O5','HNO3','HO2NO2','CO','CH2O','C
H3O2','CH3OOH','DMS','SO2','ISOP','H2SO4','SOAG','C2H5OH','CH3CHO','PS',
'lch4','r_lch4',
fincl3 = 'O3_SRF'
tropopause_e90_thrd          = 80.0e-9
history_chemdyg_summary = .false.
history_gaschmbudget = .false.
history_gaschmbudget_2D = .true.
history_gaschmbudget_2D_levels = .true.
history_UCIgaschmbudget_2D = .true.
history_UCIgaschmbudget_2D_levels = .true.
gaschmbudget_2D_L1_s =  1
gaschmbudget_2D_L1_e = 26
gaschmbudget_2D_L2_s = 27
gaschmbudget_2D_L2_e = 38
gaschmbudget_2D_L3_s = 39
gaschmbudget_2D_L3_e = 58
gaschmbudget_2D_L4_s = 59
gaschmbudget_2D_L4_e = 72
linoz_psc_t = 198.0
…
```





EOF

## Appendix B: Chemistry reactions for ozone production and loss

The following chemical reaction and loss processes are based on E3SMv3.
The chemical reaction for ozone production in chemUCI (TIP/CIP):
NO2 + hv -> NO + O3
NO3 + hv -> NO2 + O3
N2O5 + hv -> NO + O3 + NO3
OH + OH -> O3 + H2O
The chemical reaction for ozone loss in chemUCI (TIL/CIL):
O3 + H2O -> 2*OH
O3 + H2 -> OH + HO2
O3 + CH4LNZ -> OH + {CH3OO}
C2H4 + O3 -> CH2O + CO + 0.5*CH3CHO
ISOP + O3 -> MVKMACR + CH2O + OH
OH + O3 -> HO2 + O2
HO2 + O3 -> 2*O2 + OH
NO + O3 -> NO2 + O2
NO2 + O3 -> NO3 + O2
MVKMACR + O3 -> 0.5*CH3CO3 + CH2O + CH3O2
Heterogeneous reaction
The chemical reaction for ozone production suggested by Prof. Michael Prather (MPP):
CH3O2 + NO -> CH2O + HO2 + NO2
NO + HO2 -> NO2 + OH
C2H5O2 + NO -> CH3CHO + HO2 + NO2
CH3CO3 + NO -> CH3O2 + NO2
ROHO2 + NO -> CH3CHO + CH2O + NO2
ISOPO2 + NO -> MVKMACR + CH2O + NO2
MVKO2 + NO -> 0.5*CH3CO3 + CH2O + NO2
The chemical reaction for ozone loss suggested by Prof. Michael Prather (MPL):
O3 + H2O -> 2*OH
O3 + H2 -> OH + HO2
O3 + CH4LNZ -> OH + {CH3OO}
C2H4 + O3 -> CH2O + CO + 0.5*CH3CHO
ISOP + O3 -> MVKMACR + CH2O + OH
OH + O3 -> HO2 + O2
HO2 + O3 -> 2*O2 + OH
MVKMACR + O3 -> 0.5*CH3CO3 + CH2O + CH3O2

## Code and data availability:



The ChemDyg code (v0.1.4) used in this study has been released on Zenodo
(https://doi.org/10.5281/zenodo.10116320). The observational data used in ChemDyg are
available at https://doi.org/10.5281/zenodo.8274422. Other E3SMv3 simulated outputs for the
plots in this paper are available at https://doi.org/10.5281/zenodo.8415473.

**Sample availability:**
This link provides an example of a complete ChemDyg run with a testing version of E3SMv3
output
https://web.lcrc.anl.gov/public/e3sm/diagnostic_output/ac.lee1061/20220914.PAN.MZThet.v2.L
R.bi-grid.amip.chemUCI_Linozv3/e3sm_chem_diags/plots/. Other information of ChemDyg can
be found on https://e3sm.org/resources/tools/diagnostic-tools/chemdyg/.

**Author contributions:**
H.-H. Lee and all coauthors provided ideas and designed the diagnostics sets in ChemDyg. H.-H.
Lee and Q. Tang contributed to code and data development to ChemDyg. H.-H. Lee leads and
coordinates the manuscript with input from coauthors.

**Competing interests.**
The contact author has declared that none of the authors has any competing interests.

**Acknowledgements:**



This work is supported by the Energy Exascale Earth System Model (E3SM) project funded by
the U.S. Department of Energy, Office of Science, Office of Biological and Environmental
Research. Part of the study supported by the LLNL LDRD project 22-ERD-008, "Multiscale
Wildfire Simulation Framework and Remote Sensing". Work at LLNL was performed under the
auspices of the U.S. DOE by Lawrence Livermore National Laboratory under contract DE-AC52-
07NA27344. LLNL IM: LLNL-JRNL-855846-DRAFT.

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



Table 1. A summary of the ACCMIP models and UCI CTM for the surface $O_3$ comparison

| Model | Member[*] | Resolution (lat. × lon.) | Number of years | References |
|---|---|---|---|---|
| MOCAGE | r2i1p1 | 2° × 2° | 4 | Josse et al. (2004) |
| | | | | Teyssèdre et al. (2007) |
| GFDL-AF3 | r1i1p1 | 2° × 2.5° | 10 | Donner et al. (2011) |
| | | | | Naik et al. (2013) |
| CESM-CAM-SF | r1i1p1 | ~1.9° × 2.5° | 10 | Cameron-Smith et al. (2006) |
| | | | | Lamarque et al. (2013) |
| UM-CAM | r1i1p1 | 2.5° × 3.75° | 10 | Zeng et al. (2008) |
| | | | | Zeng et al. (2010) |
| CMAM | r1i1p1 | ~3.7° × 3.75° | 10 | Scinocca et al. (2008) |
| MIROC-CHEM | r1i1p1 | ~2.8° × 2.8125° | 10 | Watanabe et al. (2011) |
| GISS-E2-R | r1i1p3 | 2° × 2.5° | 5 | Koch et al. (2006) |
| | | | | Shindell et al. (2013) |
| GEOSCCM | r1i1p1 | 2° × 2.5° | 10 | Oman et al. (2011) |
| UCI CTM | - | ~2.8° × 2.8125° | 10 | Holmes et al. (2013) |

[*] The format $r <N> i <M> p <L>$ defines each model simulation's realization number (N), initialization method
(M), and perturbed physics version (L).






Table 2. A summary of the CMIP6 models the tropospheric $O_3$ comparison

| Model | Member[*] | Resolution (lat. × lon.) | Vertical layers | References |
|---|---|---|---|---|
| CESM2-WACCM | r1i1p1 | 0.9° × 1.25° | 70 | (Gettelman et al., 2019) (Tilmes et al., 2019) (Emmons et al., 2020) |
| GFDL-ESM4 | r1i1p1 | 1° × 1.25° | 49 | (Horowitz et al., 2020) (Dunne et al., 2020) |
| GISS-E2-1-G | r1i1p3 | 2° × 2.5° | 40 | (Bauer et al., 2020) |
| MRI-ESM2-0 | r1i1p1 | 2.813° × 2.813° | 80 | (Deushi and Shibata, 2011) (Yukimoto et al., 2019) |
| UKESM1-0-LL | r1i1p1 | 1.25° × 1.875° | 85 | (Archibald et al., 2020) (Mulcahy et al., 2020) |

[*] The format $r <N> i <M> p <L>$ defines each model simulation's realization number (N), initialization method
(M), and perturbed physics version (L).



Table 3. A summary of NOAA surface CO observational sites

| Code | Name | Country | Latitude | Longitude | Elevation (meters) |
|---|---|---|---|---|---|
| BRW | Barrow Atmospheric Baseline Observatory | United States | 71.323 | -156.611 | 11.0 |
| CGO | Cape Grim, Tasmania | Australia | -40.683 | 144.690 | 94.0 |
| ICE | Storhofdi, Vestmannaeyjar | Iceland | 63.400 | -20.288 | 118.0 |
| KUM | Cape Kumukahi, Hawaii | United States | 19.561 | -154.888 | 8.0 |
| MHD | Mace Head, County Galway | Ireland | 53.326 | -9.899 | 5.0 |
| MID | Sand Island, Midway | United States | 28.219 | -177.368 | 4.6 |
| PSA | Palmer Station, Antarctica | United States | -64.774 | -64.053 | 10.0 |
| RPB | Ragged Point | Barbados | 13.165 | -59.432 | 15.0 |
| SMO | Tutuila | American Samoa | -14.247 | -170.564 | 42.0 |
| SYO | Syowa Station, Antarctica | Japan | -69.013 | 39.590 | 14.0 |
| WIS | Weizmann Institute of Science at the Arava Institute, Ketura | Israel | 29.965 | 35.060 | 151.0 |
| ZEP | Ny-Alesund, Svalbard | Norway and Sweden | 78.907 | 11.888 | 474.0 |






Table 4. A summary of the basic set of diagnostics in ChemDyg

| Set of plots/tables | Description | Supported model input format | Associated observations/reanalysis data (with year range and data format) |
|---|---|---|---|
| Pressure-Latitude plots | Pressure-Latitude zonal mean contour plots of annual mean ozone (O₃), ozone in troposphere only (O₃ Trop), specific humidity (Q), and temperature (T) | Annual mean regridded (180×360) climatology | Standard E3SMv2 results (Golaz et al.) (1985-2014) |
| Latitude-Longitude plots | Latitude-Longitude contour map of annual mean vertically intergraded total precipitable water (TMQ) | Annual mean regridded (180×360) climatology | N/A |
| NOₓ emission plots | The vertical profile and Latitude-Longitude contour map of annual mean NOₓ emissions from aircraft and lightning | Annual mean regridded (180×360) climatology | N/A |
| TCO comparison | Time series global mean Tropospheric ozone (TCO) burden | Monthly regridded (180×360) time-series | CMIP6 (References see Table 2) (1985-2014) |
| Surface ozone diurnal cycle comparison | Seasonal mean hour plot of global mean surface ozone | Hourly regridded (1.0×1.0) time-series | CMIP6 (Schnell et al., 2015) (2000-2009) |
| Surface ozone annual cycle comparison | Annual mean month plot of global mean surface ozone | Hourly regridded (1.0×1.0) time - series | CMIP6 (Schnell et al., 2015) (2000-2009) |
| Surface CO comparison | Time series plot and anomalies of surface CO at NOAA ground-based facilities | Monthly regridded (180×360) time-series | NOAA surface stations (Petron et al., 2022) (1990-2020) |
| Ozone hole | Annal mean day plot of ozone hole area and minimum ozone burden in the south hemisphere | Daily time-series | Observations (https://ozonewatch.gsfc.nasa.gov/SH.html) (1990-2019) |
| Ozone hole with equivalent latitude | Time series plot and annual mean month plot of total column ozone burden with equivalent latitude | Daily time-series | N/A |
| Ozone STE flux | Time series plot and annual mean month plot of ozone stratosphere- | Monthly time-series | N/A |



| | | | |
|---|---|---|---|
| | troposphere-exchange (STE) flux | | |
| Temperature with equivalent latitude | Seasonal mean Latitude plot of temperature at specific layer with equivalent latitude | Daily time-series | N/A |
| Chemistry tendency table | Seasonal and annual mean chemistry tendency metrics summarized in tables | Annual mean climatology | N/A |
| Chemistry closure check and burden | Annual mean chemistry metrics and L2-normalization relative difference in a table | Monthly time-series | N/A |
| Chemistry production/loss tendency table | Seasonal and annual mean $O_3$ and CO production and loss tendency metrics summarized in tables | Annual mean climatology | N/A |
| Chemistry high-level summary table | Annual mean $O_3$, CO, and $CH_4$ metrics summarized in a table | Monthly time-series | N/A |


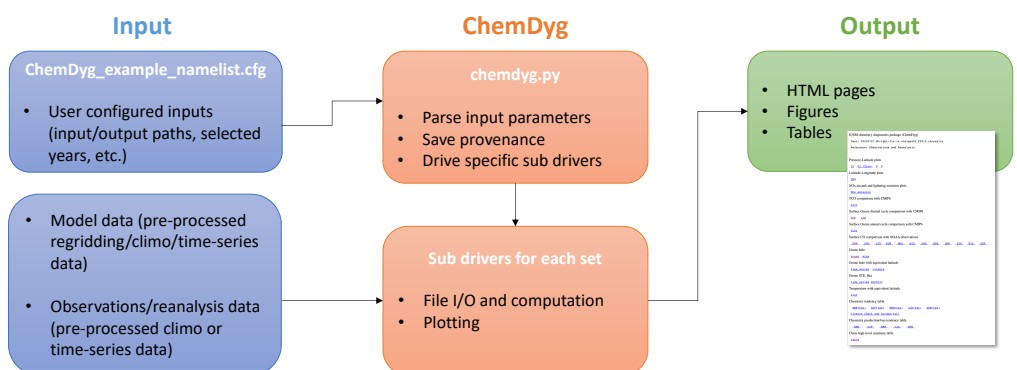

Figure 1. A schematic overview of ChemDyg structure and workflow. The primary input (blue
boxes) includes the user configuration through setting up a python configure script (i.e., the .cfg
file), model data pre-processed from native E3SM history files, and reformatted
observation/reanalysis data. The run scripts for model data pre-processing are also controlled by
the configure script. The main ChemDyg (orange boxes) parses the user input and drives
individual sub-drivers for specified diagnostics set. The output (green box) includes a HTML
page linking to each individual figures and tables.



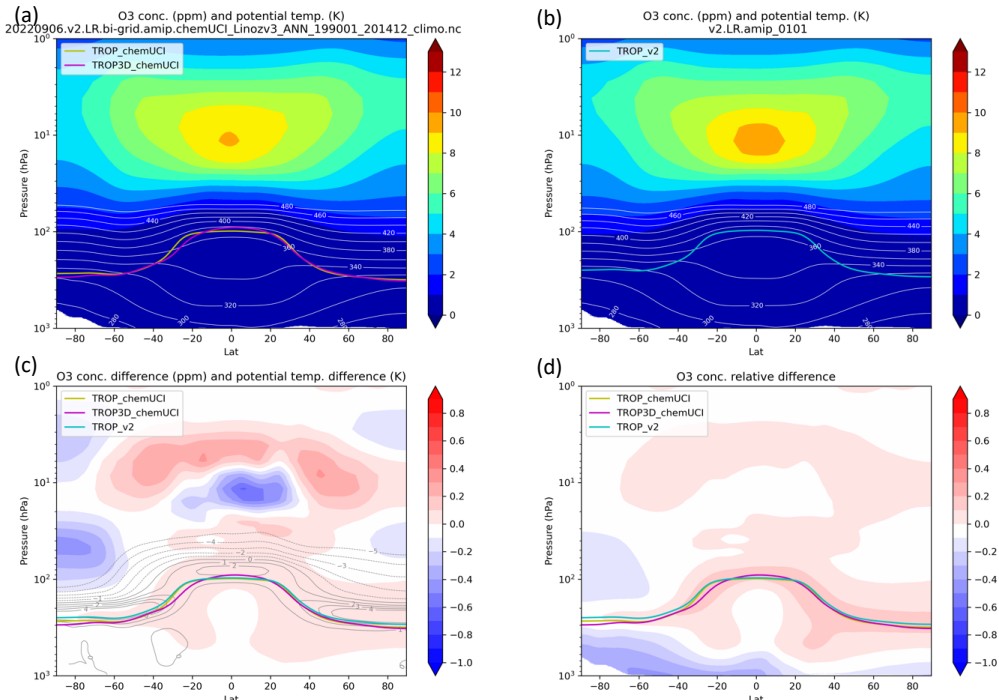

Figure 2. (a) Pressure-Latitude plot of multi-year zonal mean ozone abundance (contours; units:
ppm) and potential temperature (white lines; units: K) from a E3SMv3 testing simulation.
Yellow line and magenta line show the WMO defined tropopause and 3D tropopause,
respectively. (b) is same as (a) but for the standard E3SMv2 simulation. Cyan line shows the
WMO defined tropopause. (c) and (d) are the absolute difference and the relative difference of
(a) and (b), respectively.

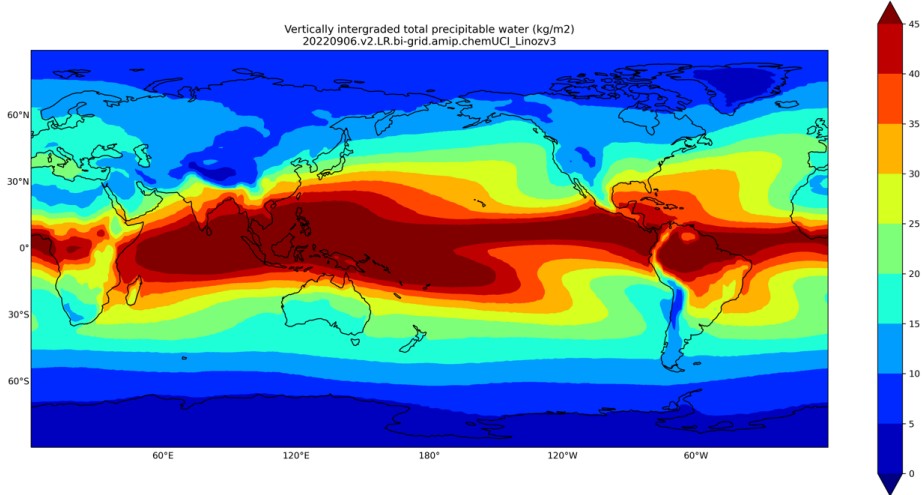

Figure 3. Latitude-Longitude contour map of annual mean vertically intergraded total
precipitable water (TMQ; units: kg/m$^2$) from a E3SMv3 testing simulation.



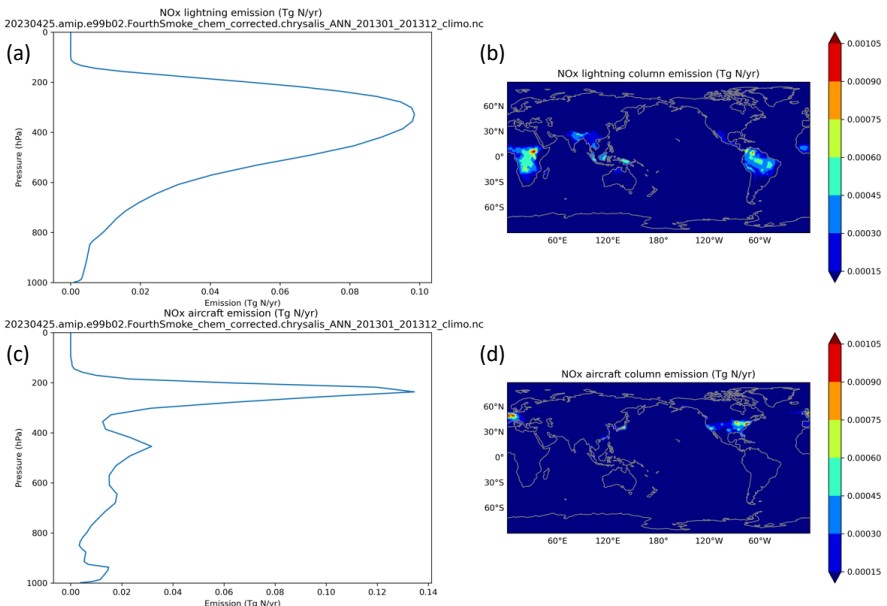

Figure 4. (a) The vertical profile of global sum lightning discharged $NO_x$ (units: Tg N/year) and
(b) Latitude-Longitude contour map of annual mean vertically intergraded lighting $NO_x$ (units:
Tg N/year) from a E3SMv3 testing simulation. (c) and (d) are the same as (a) and (b) but for
aircraft emitted $NO_x$.




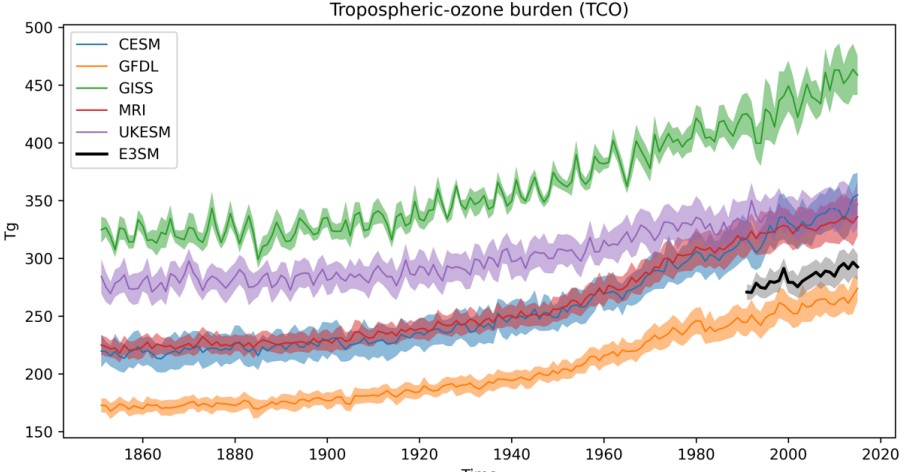


Figure 5. Time series of global mean tropospheric column ozone (TCO) burden (units: Tg) in 5
CMIP6 models listed on Table 2 and E3SM (black line). The shading shows the mean ± 1
standard deviation of the monthly variability for each year.




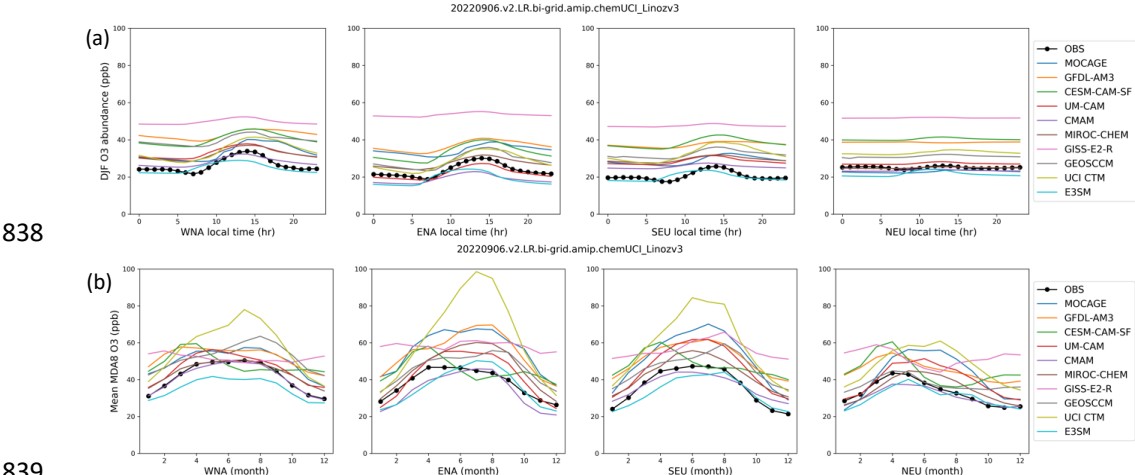


Figure 6. (a) Diurnal cycles of hourly O$_3$ abundance (units: ppt) for observations (OBS; black
lines), 8 ACCMIP models listed on Table 1, UCI CTM (yellow-green lines), and E3SM (cyan
lines) averaged over winter (DJF) months in WNA, ENA, SEU, and NEU from left panel to
right, respectively. (b) Annual cycles of MDA8 O$_3$ abundance (units: ppt) for observations
(OBS; black lines), 8 ACCMIP models listed on Table 1, UCI CTM (yellow-green lines), and
E3SM (cyan lines) in WNA, ENA, SEU, and NEU from left panel to right, respectively.



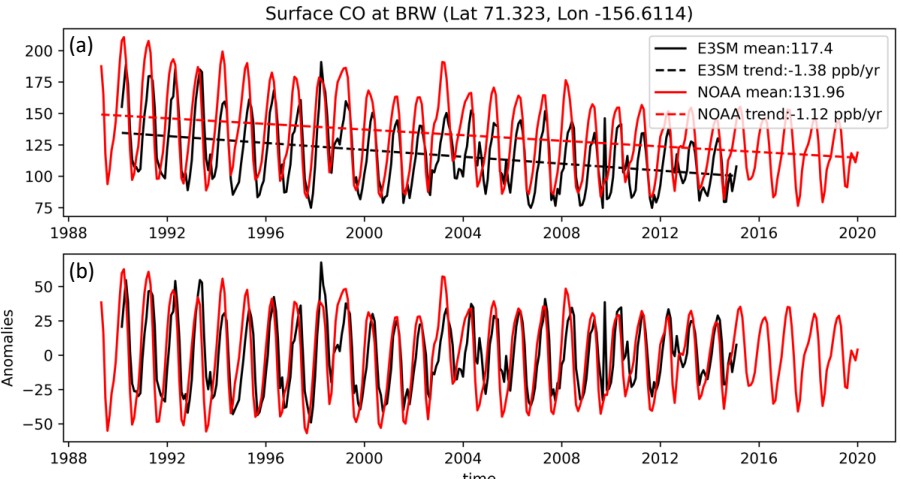

Figure 7. (a) Time series of surface CO burden (units: ppb) from observation (red solid line) and
E3SM (black solid line) at the BRW observational site listed on Table 3. Red dotted line and
black dotted line show the tendency of surface CO burden of observation and E3SM,
respectively. (b) Evolution of anomalies of surface CO burden from observation (red solid line)
and E3SM (black solid line).

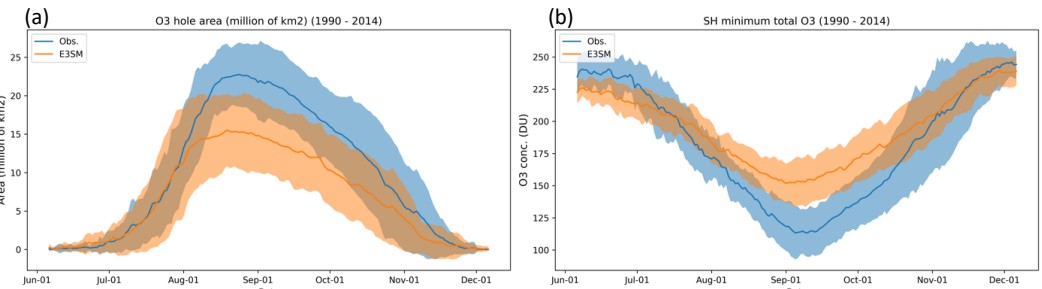

Figure 8. Daily evolution of the Antarctic ozone hole from 1 July to 31 December as measured by (a) area ($10^6$ km$^2$) and (b) minimum total column ozone (DU). Results are shown for the model (orange lines) and observations (blue lines) from the NASA ozone watch data for 1990-2019. The shaded area covers ± 1 standard deviation of the multi-year variability for each day.

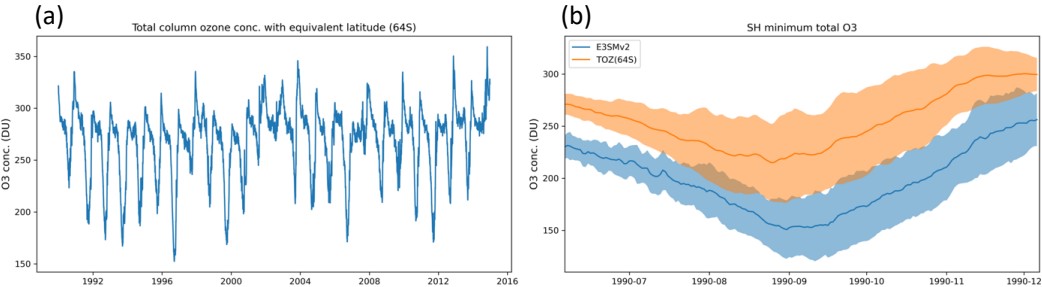

Figure 9. (a) Time series of the minimum total column ozone (DU) with equivalent latitude (64S) area from 1 July to 31 December over the years 1990-2014. (b) Daily evolution of the Antarctic ozone hole from 1 July to 31 December as measured by the minimum total column ozone (DU). Results are shown for the model (blue line) and derived from (a) (orange line). The shaded area covers ± 1 standard deviation of the multi-year variability for each day.

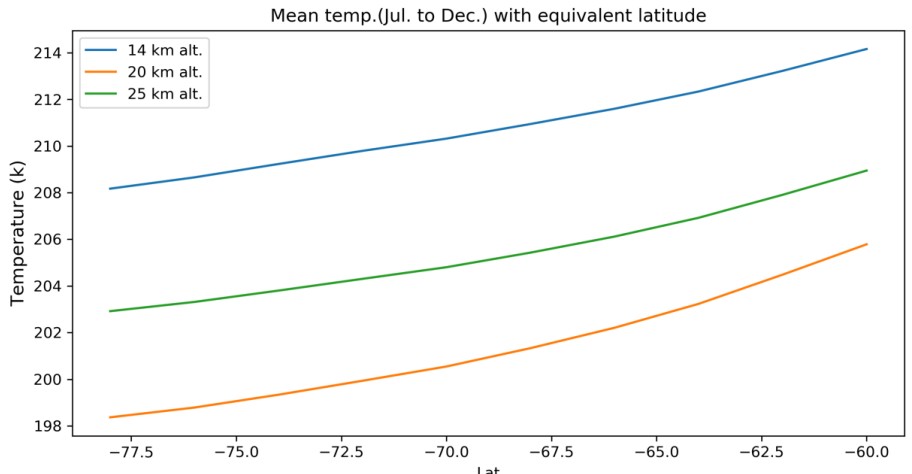

Figure 10. Mean temperature averaged from 1 July to 31 December over the years 1990-1999 at 14, 20 and 25 km altitude with equivalent latitudes from 60°S to 78°S.




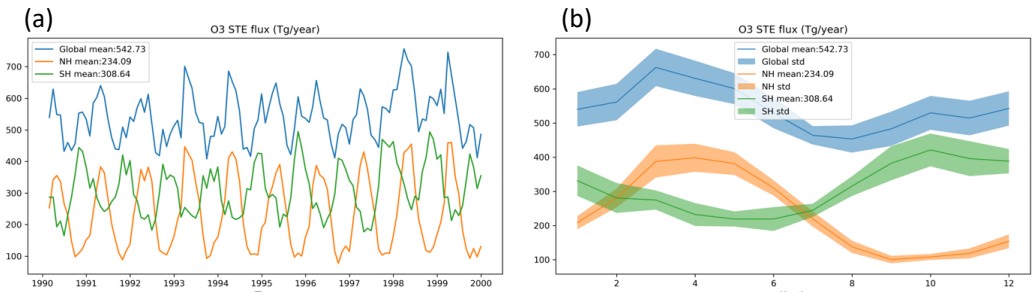

Figure 11. (a) Time series of the stratosphere troposphere exchange (STE) ozone fluxes (unit:
Tg/year) over the years 1990-2000. (b) Mean annual cycle of STE ozone fluxes in the north
hemisphere (NH; orange line), south hemisphere (SH; green line) and global (blue line). The
shaded area covers ± 1 standard deviation of the multi-year variability for each month.





Figure 12. A schematic workflow of chemistry tendency processes and their associated working
files.






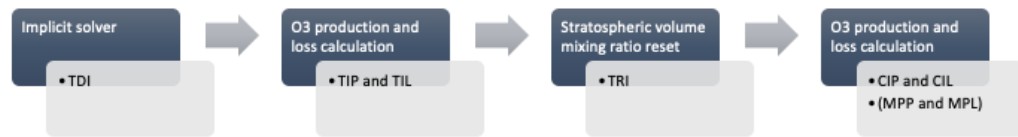

Figure 13. A schematic workflow of chemistry tendency processes for the ozone production and
loss diagnostics in the implicit solver and after stratospheric reset.