# Peer review of "E3SM Chemistry Diagnostics Package (ChemDyg) Version 0.1.4"

_Geoscientific Model Development, 2023_

## Author Comment (AC1)

"E3SM Chemistry Diagnostics Package (ChemDyg) Verions 0.1.4" [10.5194/gmd-2023-203]

Responses to the Comments of the Anonymous Referee #1

We very much appreciate the constructive comments and suggestions from this reviewer. Our point-by-point responses to the reviewer's comments are as follows (the reviewer's comments are marked in Italic fort).

**General Comments:**

*This paper is a description of the ChemDyg diagnostic package v 0.1.4 capable of evaluating the novel model chemistry improvements made in version 2 of the E3SM model. The authors describe the collection of observational datasets and other model data used for such evaluation.*

*The overall comment is that it is not clear what the scientific contribution of this paper is. A few ways in which the authors could highlight this are:*

*1. It would be helpful if the authors could emphasize the novelty in the evaluation, as in what the evaluation package does that cannot be done by other model evaluation tools. There is no comparison with any other evaluation tool except where they state upgrades from a previous version.*

We value the reviewer's input. In the revised manuscript, we have underscored distinct diagnostic features within ChemDyg. We've also incorporated additional novel diagnostic plots, such as the spatiotemporal pattern of stratosphere-troposphere exchange (STE) flux of ozone and quasi-biennial oscillation (QBO)-ozone diagnostics. Our updated version of ChemDyg (v1.0.1) boasts highlights high temporal resolution data on ozone hole and temperature variation with equivalent latitude, along with detailed chemistry tendency tables for each chemical process. Approximately half of the plots and tables in ChemDyg are routinely utilized for model evaluation, while the remaining half are tailored for specific scientific applications, providing unique insights that are not readily available in other diagnostic tools.

*2. It is not clear if there is a broader applicability to the tool. Can it be used with other models (not just for comparison with E3SM)? If yes, some examples will be useful. If it cannot be applied more broadly, then does it only serve the purpose of a monitoring tool for the new chemistry in E3SM? That would be very specific in application.*

As stated at the beginning of the paper, ChemDyg is designed to support the Department of Energy (DOE) Energy Exascale Earth System Model (E3SM). Its direct application to other models will require substantial additional effort to address issues such as discrepancies in output format or variable names. Such effort would not be supported by the E3SM funds because it's beyond the E3SM scope. Nonetheless, being an open-source software, users have the flexibility to adapt the source codes as examples to suit their specific requirements for model evaluation and specialized diagnostics. In that sense, the application of ChemDyg can be potentially

broader. The chemistry tendency table and tendency closure check were specifically tailored for E3SM model developers and may not be readily compatible with other models. We have clarified this aspect in the revised manuscript. Should funding resources become available to make ChemDyg compatible with other models outputs, we would be happy to do so in the future.

*3. The plots in Figures 2 onwards are standard evaluation plots. It is not clear that there are new metrics or diagnostics in the evaluation. So, again the significance of the evaluation is not clear. It seems the scientific novelty is still in the new chemistry package and not in the evaluation.*

See the reply to the general comment #1 above.

**Specific comments:**

*1. In Line 92, the authors say that the ozone only diagnostics cannot meet the EAMv3 chemistry evaluation requirements. It is not explained why this is so.*

We modified the manuscript to "The ozone-only diagnostics in E3SM Diags v2.7 cannot meet the EAMv3 chemistry evaluation requirements.  For example, E3SM Diags only includes TCO and SCO zonal mean annual cycle.  However, we need a systematized analysis tool for E3SM chemistry to extend model evaluation for ozone hole and other specific diagnostics, like STE flux of ozone.  Thus, ChemDyg is necessary for E3SM model developers and users."

*2. Where examples of diagnostics are provided in Section 3, it would be useful if the authors can highlight the importance of the evaluation with suitable references. This would be a way to demonstrate the scientific value of the paper.*

Thank you for your valuable suggestion. We appreciate your insight into highlighting the importance of evaluation with suitable references in Section 3. We carefully consider incorporating references that underscore the scientific significance of the diagnostics presented. By providing detailed explanations and referencing relevant literature, we aim to enhance the overall scientific value of the paper.

**Technical comments:**

*1. Line 27: dimentional -> dimensional*

Similar errors are modified through the whole manuscript.

*2. Line 46: farm ->frame*

Modified.

*3. Line 389: summery -> summary*

Modified.

*The paper seems like a technical documentation of the ChemDyg diagnostic package but for it to be a scientific publication the above comments have to be addressed at the very least.*

---

## Author Comment (AC2)

**"E3SM Chemistry Diagnostics Package (ChemDyg) Verions 0.1.4" [10.5194/gmd-2023-203]**

Responses to the Comments of the Anonymous Referee #2

We very much appreciate the constructive comments and suggestions from this reviewer. Our point-by-point responses to the reviewer's comments are as follows (the reviewer's comments are marked in Italic fort).

**General Comments:**

*This manuscript discusses the E3SM Chemistry diagnostics package (ChemDyg), which is open-source software written in Python. It was developed to support the Department of Energy's Energy Exascale Earth System Model (E3SM). ChemDyg version 0.1.4 generates various diagnostic plots and tables (11 types of plots and 4 types of tables) for comparing model outputs to both other models and observational data. These include 2-dimensional contour mapping plots, diurnal and annual cycle plots, time-series plots, and comprehensive processing tables. ChemDyg is executed by zppy, a post-processing toolchain for E3SM written in Python. The codebase of ChemDyg is modular, with each diagnostics set being self-contained. Each set has its own driving script for file input/output and a main Python script for calculation and plotting. The outputs, including figures and tables, are organized in an interactive HTML page accessible through a browser.*

*The manuscript is well structured and written. It provides a detailed description of E3SM ChemDyg version 0.1.4, including the specifics of each diagnostics set and the required input data formats. It also highlights the tool's flexibility for future developers to incorporate new observational datasets and diagnostics sets.*

*The manuscript lacks direct comparisons with existing tools. A paragraph on the tool's novelty and added value compared to other evaluation tools is essential. Highlighting specific features or functionalities that distinguish ChemDyg from other evaluation tools would provide readers with a clearer understanding of its unique contributions to model evaluation tasks.*

Thanks for the suggestion. As the reply to the reviewer 1, in the revised manuscript, we have underscored the distinctive diagnostic features within ChemDyg. We've also incorporated additional novel diagnostic plots, such as the spatiotemporal pattern of stratosphere-troposphere exchange (STE) flux of ozone and quasi-biennial oscillation (QBO)-ozone diagnostics. Our updated version of ChemDyg (v1.0.1) boasts high temporal resolution data on ozone hole and temperature variation with equivalent latitude, along with detailed chemistry tendency tables for each chemical process. Approximately half of the plots and tables in ChemDyg are utilized for model evaluation, while the remaining half are tailored for specific scientific contributions, providing unique insights that other diagnostic tools struggle to replicate.